# Influence of Thermal Treatment and Acetic Acid Concentration on the Electroactive Properties of Chitosan/PVA-Based Micro- and Nanofibers

**DOI:** 10.3390/polym15183719

**Published:** 2023-09-10

**Authors:** Rigel Antonio Olvera Bernal, Roman O. Olekhnovich, Mayya V. Uspenskaya

**Affiliations:** Chemical Engineering Centre, ITMO University, Kronverkskiy Prospekt, 49A, St. Petersburg 197101, Russia

**Keywords:** electroactive polymers, chitosan, poly (vinyl alcohol), acetic acid, soft actuators

## Abstract

This study presents, for the first time, a comprehensive investigation of the influence of pre- and post-fabrication parameters for the electroactive properties of electrospun chitosan/PVA-based micro- and nanofibers. Chitosan/PVA fibers were fabricated using electrospinning, characterized, and tested as electroactive materials. Solutions with different acetic acid contents (50, 60, 70, and 80 *v/v*%) were used, and the rheological properties of the solutions were analyzed. Characterization techniques, such as rheology, conductivity, optical microscopy, a thermogravimetric analysis, differential scanning calorimetry, a tensile test, and FT-IR spectroscopy, were utilized. Fiber mats from the various solutions were thermally treated, and their electroactive behavior was examined under a constant electric potential (10 V) at different pHs (2–13). The results showed that fibers electrospun from 80% acetic acid had a lower electroactive response and dissolved quickly. However, thermal treatment improved the stability and electroactive response of all fiber samples, particularly the ones spun with 80% acetic acid, which exhibited a significant increase in speed displacement from 0 cm^−1^ (non-thermally treated) to 1.372 cm^−1^ (thermally treated) at a pH of 3. This study sheds light on the influence of pre- and post-fabrication parameters on the electroactive properties of chitosan/PVA fibers, offering valuable insights for the development of electroactive materials in various applications.

## 1. Introduction

Soft actuators are systems inspired by muscles in human and animals, which present characteristics such as agility, environmental adaptability, flexibility, and multifunctionality, among others [1]. Soft actuators have potential uses in soft robotics, artificial muscles, wearable devices, biomedicine, soft grippers, and so forth [2,3,4,5]. Mainly, soft actuators are fabricated using soft materials [6] and can be driven by different physical and chemical stimuli (heat, magnetic fields, humidity, fluid pressure, pH variations, electric fields, and so forth) [7,8,9,10]. Among these, electrically driven materials, such as electroactive polymers (EAPs), have gained attention in that they resemble natural muscles [11,12]. According to the actuation mechanism, EAPs are grouped into two broad categories. When coulomb forces govern the actuation mechanism, they are referred to as electronic EAPs. Within this group, we can find ferroelectric polymers, electrostrictive graft elastomers, dielectric elastomers, liquid crystal elastomers, and electroviscoelastic elastomers. On the other hand, when the ions’ diffusion or mobility causes the actuation mechanism, they are referred to as ionic EAPs. Herein, we can find conductive polymers, ionic polymer gels, and ionic polymer–metal composites (IPMCs) [13,14]. The electroactive properties of a polymer can be described as a mechanical response to an electrical stimulus, or in other words, the ability of a polymer to deform by changing its shape or size when excited by electrical potentials [11,12,13,14].

In recent years, extensive scientific research has focused on the potential application of micro- and nanofibers as electroresponsive hydrogels. Despite being a relatively new material, their convergence of favorable properties, such as flexibility, elasticity, substantial porosity, adjustability, and other characteristics, has shown great promise in the context of electroactive polymers (EAPs) [15]. Nanofibers are highly desirable as electroresponsive materials due to their unique morphology, which offers several advantages. One significant advantage is the shortened response time achieved by reducing the size of the hydrogel, as the actuation rate is inversely proportional to its size. Additionally, nanofibers exhibit a one-dimensional structure with a significant surface-to-volume ratio, leading to the greater ionization of functional groups when interacting with the surrounding medium. This is primarily attributed to the larger surface area of each individual fiber. Furthermore, the high porosity of nanofiber mats effectively facilitates the diffusion of free ions, allowing them to migrate between a material’s interior and exterior. This enhanced ion diffusion contributes to greater material deformation and improved mechanical strength compared to that of hydrogels [16].

The demand for the development of polymeric micro- and nanofibers has increased over the past decade, due to their potential use in various applications, such as protective clothing, tissue engineering, drug delivery systems, energy storage, filtration, functional materials, sensors, and soft actuators [17]. For the production of micro- and nanoscale fibers, several techniques have been developed. Based on the fiber formation process, these techniques can be divided into two groups. The first of these, in which the fibers’ production is due to electrostatic forces, are called electrospinning techniques [18]. The second group of techniques use mechanical forces for fiber formation, such as drawing, template synthesis, phase separation, and so forth [19]. Electrospinning is a mechanical and electrical technique that allows for the production of submicrometric fibers [20,21]. This technique makes it possible to fabricate hierarchical structures that can emulate the fibrous structure of biological muscles, which is a desired characteristic when designing soft actuators [22]. Electrospinning involves applying a high voltage to a polymer solution as it is fed through a capillary or needle. This process results in the formation of a conical jet, known as a Taylor cone, due to the accumulation of an electric charge at the tip of the droplet. Subsequently, the thin yarn is gathered onto a grounded electrode plate. Electrospun fiber morphology is influenced by numerous factors, including technical parameters such as flow, voltage, and needle-to-collector distance; solution properties, such as conductivity, viscosity, and polymer concentration; and environmental conditions, such as temperature and humidity [23].

In recent decades, the use of biopolymers as electroactive materials has been widely explored. Chitosan, cellulose, starch, and alginate, among others, have shown good electroactive properties due to the presence of polar groups that are easily polarizable [24,25]. Chitosan is a copolymer composed of *N*-acetyl-d-glucosamine and d-glucosamine. The degree of deacetylation (DDA) achieved during the conversion of chitin to chitosan determines the ratio of d-glucosamine in the copolymer. A variety of organic and inorganic acids have been found to be effective solvents for chitosan, such as malic acid, l-glutamic acid, acetic acid, formic acid, and hydrochloric acid, among others [26,27]. Acetic acid and trifluoroacetic acid (TFA) are the most commonly used solvents for electrospinning chitosan, especially for achieving uniform fiber formation. Additionally, acetic acid influences the rheological properties of chitosan solutions, which affect the morphology of electrospun fibers [28,29]. Chitosan possesses amino groups and hydroxyl groups along its backbone, resulting in its polycationic nature. This polycationic property has led to extensive research on chitosan for the advancement of electroresponsive materials [30,31]. Zolfagharian A. et al. [32] developed a 3D-printed, soft, biodegradable chitosan-based hydrogel actuator. Shamsudeen K. et al. [33] investigated anionic and cationic polyelectrolyte hydrogels’ electroactive response. Partially hydrolyzed polyacrylamide/PVA was synthesized as an anionic hydrogel; meanwhile, chitosan/PVA was synthesized as a cationic hydrogel. The cationic and anionic gels exhibited contrasting behaviors which have various applications in areas such as medicine and robotics.

The formation of pure chitosan fibers by the electrospinning process has proven to be a challenging task due to the high electrostatic repulsion forces between chitosan chains. To overcome this problem, chitosan is often mixed with a highly electrospinnable polymer, such as poly(ethylene oxide) or poly(vinyl alcohol), which neutralizes the repulsive forces [34,35,36]. The latter is the most used polymer to blend with chitosan. Polyvinyl alcohol (PVA) is a partially crystalline polymer with a linear structure. This consists of a carbon backbone and —OH groups. PVA possesses several significant characteristics, such as its wide availability, solubility in water, excellent film-forming properties, and thermal stability, among others [37,38].

In our previous study, we made a significant contribution by introducing the utilization of biopolymer nanofibers for the development of a fibrous, electroresponsive soft actuator based on electrospun chitosan/PVA nanofibers [39]. To the best of the authors’ knowledge, this innovative approach had not been previously reported in the scientific literature. In our previous work, we aimed to study the influence that chitosan has on the electroactive behavior of materials. For this aim, chitosan/PVA fibers were fabricated with different chitosan concentrations and dissolved in a 40% aqueous acetic acid solution. The obtained results showed that the fibers with a higher content of chitosan had faster bending displacement in response to an electric stimulus. Other studies have used nanofibers for the development of electroresponsive soft actuators as well, such as those reported by Riccardo D’Anniballe et al. [40], Asai H. et al. [41], Seyed V. et al. [42], Miranda D. et al. [43], and Ismail Y. et al. [44]. Many of these works focus on how the electroactive behavior of fibrous soft actuators can be influenced by the type of polymers utilized, the differential potential, the conductivity of the material, and the electrolyte solution.

Nevertheless, as of the writing of this article and to the best of the authors’ knowledge, no previous reports have explored the influence of pre- and post-fabrication parameters on the electroresponsive behavior of fibrous soft actuators based on electrospun fibers. This knowledge gap highlights the novelty and significance of our current study as we aim to address this research gap and shed light on the crucial factors that impact the electroresponse of electrospun fibers. 

## 2. Materials and Methods

### 2.1. Materials

Chitosan powder with a molecular weight of 260 kDa and PVA powder with a molecular weight of 74 kDa were purchased from the company “Bioprogres” (Moscow, Russian Federation). Acetic acid (99.5%) and distilled water were used as components of the binary solvent system. All components were used as received.

### 2.2. Chitosan and Chitosan/PVA Solutions Preparation

For conductivity and viscosity analysis, solutions containing only chitosan and chitosan/PVA were made by dissolving the polymers in a binary solvent system composed of distilled water and acetic acid at different concentrations. All polymeric solutions were stirred in a hot plate magnetic stirrer (IKA Magnetic stirrer RH, IKA, Staufen, Germany).

To prepare the chitosan solutions, the polymeric concentration was kept constant at 4 wt% and dissolved in acetic acid/distilled water binary solvent system at different ratios (50:50, 60:40, 70:30, and 80:20 acetic acid:distilled water, respectively). The solutions were mixed at a temperature of 90 °C until the solutions were homogenous. The solutions were labeled as CsAA50, CsAA60, CsAA70, and CsAA80, respectively.

To prepare the chitosan/PVA solutions, firstly, chitosan (8 wt%) was dissolved in a binary solvent composed of distilled water and acetic acid at different concentrations (50, 60, 70, and 80%) at a temperature of 90 °C. Afterwards, PVA solutions (10 wt%) were prepared by dissolving PVA powder in a binary solvent composed of distilled water and acetic acid at different concentrations (50, 60, 70, and 80%) at a temperature of 80 °C. Once all solutions were homogenous, chitosan and PVA solutions were mixed in a volume ratio of 1:1 in order to prepare solutions with constant polymeric concentrations while maintaining a varying acetic acid concentration. The chitosan/PVA concentrations of the solutions were as follows: chitosan 4%/PVA 5% dissolved at different acetic acid concentrations (50, 60, 70, and 80%). These solutions were also used as precursor solution for electrospinning and the formation of fibers. The solutions were labeled as AA50, AA60, AA70, and AA80, respectively. 

### 2.3. Rheological Properties and Conductivity of Polymeric Solutions

The polymeric solutions were characterized by rotational viscometry using an MCR 502 rheometer (Anton Paar, Graz, Austria) equipped with a C-PTD 200 temperature control module and a CC27 cylinder-bowl measuring system (ISO 3219-1:2021). The solutions were measured at a constant temperature of 25 °C and over a shear rate range of 0.1–500 s^−1^ with a logarithmic profile for shear rate variation.

The conductivity of the polymeric solutions was measured on a SevenCompact Duo S213-meter, pH/Ion dual channel benchtop meter, with a conductivity probe sensor (InLab^®^ 738, Meter Toledo, Viroflay, France).

### 2.4. Chitosan/PVA Fiber Production by Electrospinning Method

The fibers were produced by electrospinning the chitosan/PVA solutions containing different acetic acid concentrations. The electrospinning process was performed in an NANON-01A system (MECC CO., LTD, Fukuoka, Japan). The fiber mats were collected on a rotational drum (with a rotation speed of 500 rpm) at 26 ± 2 °C and a relative humidity of 23 ± 1%. All fiber mats were collected on aluminum foil, after the polymeric solution was electrospun for 13 h. A total volume of 5 mL was used for the obtention of the fiber mats. For the electrospun chitosan/PVA fibers, the electrospinning technical values, such as the voltage (30 kV), needle–collector distance (150 mm), and feed rate (0.2 mL/h), were the same for all samples. The fibers mats obtained from the polymeric precursor’s solution were labelled as follows: F-AA50 (chitosan 4%/PVA 5% dissolved in 50% acetic acid); F-AA60 (chitosan 4%/PVA 5% dissolved in 60% acetic acid); F-AA70 (chitosan 4%/PVA 5% dissolved in 70% acetic acid); and F-AA80 (chitosan 4%/PVA 5% dissolved in 80% acetic acid).

### 2.5. Thermal Treatment

The thermal treatment was conducted in an electronically controlled drying oven with natural convection (Binder ED 53, BINDER GmbH, Tuttlingen, Germany). The electrospun chitosan/PVA fiber mats were heated at a temperature of 70 °C, with a temperature variation of ±2 °C (in accordance with the equipment data sheet) for a period of 24 h. All samples were hermetically sealed in a resealable storage bag after thermal treatment until further use.

### 2.6. Morphology and Diameter Determination of Chitosan/PVA Electrospun Fibers

An Olympus STM6 optical microscope (OLYMPUS Corporation, Tokyo, Japan) was used to evaluate the morphology of electrospun chitosan/PVA fibers. Differential interference contrasting technique (DIC) was utilized to emphasize the colorfulness and contrast of the micrographs.

The micrographs obtained were used to analyze and measure the diameter of electrospun fibers using ImageJ 1.53e software (National Institutes of Health, Bethesda, MD, USA). A total of 200 fibers from each sample were measured to calculate the average diameter of the fibers obtained.

### 2.7. Fourier-Transform Infrared Spectroscopy FTIR Analysis

All the electrospun chitosan/PVA fibers were analyzed before and after thermal treatment. All samples were analyzed using a Bruker Tensor 37 Fourier-transform infrared (FTIR) spectrometer (Bruker, Billerica, MA, USA) over a wavenumber range of 4000–600 cm^−1^ with a resolution of 2 cm^−1^ and after 32 scans.

### 2.8. Thermogravimetric and Differential Scanning Calorimetry Analysis

The thermal properties of the obtained chitosan/PVA fibers were analyzed before and after thermal treatment. TG 209 F1 Libra (Netzsch, Berlin, Germany) was used for the thermogravimetric analysis (TGA). For the study, fiber samples weighing 6 mg were prepared. The analysis was performed in a temperature range from 25 °C to 600 °C with a gradual rise of 10 K/min, under a nitrogen atmosphere with a flow rate of 40 mL/min.

For the differential scanning calorimetry analysis, chitosan/PVA fiber samples, sealed in aluminum pans, were subjected to two heating cycles. The first heating cycle aimed to eliminate any residual water or solvent traces present in the samples. The heating cycles were as follows. First, the cycle started at 25 °C, rising to 150 °C, then the temperature was decreased to −30 °C. Subsequently, the second cycle started at −30 °C and concluded at 280 °C. The analysis was performed using an NETZSCH DSC 204F1 Phoenix under a nitrogen atmosphere. The analyzed sample weighed around 4 mg. The software OriginPro 2018 SR1 (OriginLab Corporation, Northampton, MA, USA) was utilized to calculate the area of PVA melting peak. The following equation describes the calculation for the degree of crystallinity of PVA.
(1)$$x_{c}=\frac{∆H}{∆H_{c}}\times 100$$

Here, ∆H is the sample’s enthalpy of melting and $∆H_{c}$ denotes the melting enthalpy of 100% crystalline PVA, which is assumed to be 138.6 J/g [45,46].

### 2.9. Tensile Properties of Electrospun Fibers

Chitosan/PVA electrospun fibers’ tensile properties were evaluated before and after thermal treatment. The tensile test was performed according the ISO 527-3 standard. A tensile testing machine (Instron 5943, Instron, Norwood, MA, USA) was used to obtain the stress–strain curves for the tensile tests of the electrospun fibers. From the obtained curves, Young’s modulus, tensile strength, and elongation at break were evaluated. Four samples of each composition were tested. Rectangular samples were cut to a size of 120 × 10 mm (length × width) and had a thickness of 32, 42, 39, and 34 µm for F-AA50, F-AA60, F-AA70, and F-AA80 (before heat treatment), respectively, and a thickness of 27, 21, 22, and 31 µm for F-AA50, F-AA60, F-AA70, and F-AA80 (after heat treatment), respectively. All tensile tests were conducted at room temperature using a testing speed of 10 mm/min and a distance of 100 mm between grips. The sample thickness was determined by calculating the average of three measurements taken at different points across all samples. A digital micrometer (Techrim T050011, TEXPИM, Russian Federation) with an error of ±0.003 mm (in accordance with the equipment data sheet) was used to measure the thickness of the fiber samples.

### 2.10. Electroactive Test Response

Figure 1 shows a schematic diagram of the system used for the study of the fibers’ bending displacement in response to an electric stimulus. The electroactive response tests for both thermally and non-thermally treated fibers were performed in an electrochemical cell. In order to examine the chitosan/PVA fibers’ electroactive behavior, samples obtained from the fiber mats were placed in both acidic and basic solutions (pHs of 2–12) in between two titanium electrodes. Samples with dimensions of 20 × 3 mm (length × width) were used in this experiment.

To measure the speed displacement of the fibers, a digital video camera recording at 30 FPS was used to record the electromechanical response of the material. The polarity of the electrodes was reversed at 1 s intervals, controlled by a microcontroller (Arduino nano) connected to the L298N driver. A total of three minutes of electrical stimulation were applied to the fibers. From the obtained videos, frames from different time periods were selected. The chosen images were exported to the ImageJ software platform, wherein the linear displacement was quantified in relation to time. The indicated rate of displacement (mm s^−1^) was computed as the mean of multiple measurements conducted at various time intervals within the video. A total of 5 measurements were taken to derive the average value.

## 3. Results and Discussion

### 3.1. Rheological Properties and Conductivity of Polymeric Solutions

During the electrospinning process, the rheological properties of polymer solutions are critical to fiber formation. Fiber morphology is significantly influenced by parameters such as viscosity and conductivity. Section 3.2 (Morphology and Diameter Distribution of Chitosan/PVA Nanofibers) demonstrates this correlation effectively. All the measured solution exhibited non-Newtonian behavior. Shear thinning or pseudoplastic flow was observed, possibly due to the ionized —NH_2_ groups in chitosan [47,48,49]. The viscosity is notably increased as a result of the hydrogen bonding between chitosan’s —NH_2_ and —OH groups with the —OH groups of PVA, polymer–polymer entanglement, and randomly orientated polymeric chains. As the shear rate goes higher, the viscosity decreases, which can be attributed to the exposed —NH^3^ groups affecting the electrostatic and steric repulsion and the realigned polymeric structure [50,51].

Firstly, the viscosity of polymeric solutions obtained by the dissolution of chitosan in different concentrations of acetic acid (50, 60, 70, and 80%) was studied. Figure 2 shows the viscosity dependency in relation to the concentration of acetic acid in chitosan solutions CsAA50, CsAA60, CsAA70, and CsAA80 at room temperature. From the obtained results, it is noticeable that, as the acetic acid concentration increases, the chitosan solution’s viscosity increases. The apparent viscosity measured at $\dot{\gamma }$ = 50 s^−1^ increased from 986.93 mPa·s (CsAA50) to 2014.2 mPa·s (CsAA70). The protonation of the amino groups may explain this behavior. In an acidic medium, —NH_2_ gets protonated (—NH_3_^+^), and the increase in charge density throughout the molecular chain results in the unfolding of the chain. The degree of entanglement and intermolecular interaction between the polymer chains increases as the polymer chain unfolds. Therefore, as the concentration of acetic acid is higher, more hydrogen bonds are formed between chitosan and the acetic acid, causing the viscosity of the solutions to increase. Nonetheless, it was observed that the viscosity was drastically reduced at an acetic acid concentration of 80% (1262.8 mPa·s).

Afterwards, we evaluated the viscosity of polymeric solutions containing chitosan/PVA dissolved in acetic acid at various concentrations (50%, 60%, 70%, and 80%). The results clearly indicate that the inclusion of poly (vinyl alcohol) causes a significant increase in the polymer solutions’ viscosity, as depicted in Table 1. Furthermore, as shown in Figure 2, the viscosity of the solutions increases as the concentration of acetic acid increases. In contrast to the chitosan solutions, which showed varying viscosity trends, the chitosan/PVA solutions exhibited a consistent increase in viscosity as the concentration of acetic acid increased, ranging from 6157 mPa·s (AA50) to 8747.3 mPa·s (AA80) measured at $\dot{\gamma }$ = 50 s^−1^. This phenomenon arises due to the entanglement of the polymer chains and the formation of numerous hydrogen bonds between the —H groups present in PVA and the ionized amino groups and —OH groups present in chitosan. 

The decrease in viscosity for the chitosan solution with 80% acetic acid can be explained as follows. The decrease in viscosity with the increasing electrolyte concentration can be explained by the shielding effect of counterions. Due to ionic dipole forces, acetate ions form a cascade of negatively charged particles over each chitosan molecule, creating Coulomb repulsion forces between them. This leads to a decrease in the flow resistance [29,52]. As it was reported by Kienzle-Sterzer et al. [53] in a solution with a high acetic acid content, another possible explanation for this behavior is that some of the protonated ions located in the chitosan structure are neutralized by the increase in acetate ions. Due to this neutralization, the flexibility of the polymer chain increases, causing the macroion domain to shrink, thus decreasing the viscosity.

Another property that changes depending on the content of acetic acid in the polymeric solution is the conductivity. From the obtained results, it is possible to notice that the conductivity is affected by two factors: the addition of PVA to the chitosan solutions and the acetic acid concentration, as shown in Table 2. The conductivity decreases with the increasing acetic acid concentration, which may be related to the increase in the density and strength of hydrogen bonds, which reduce the number of free charged groups. In addition, when PVA is combined with chitosan, the formation of both intra- and intermolecular bonds reduces the number of free charged groups in the polymer chain.

### 3.2. Morphology and Diameter Distribution of Chitosan/PVA Electrospun Fibers

The electrospinning process was conducted under environmental conditions at 26 °C and an RH of 23%. The fibers were electrospun from solutions composed of chitosan/PVA dissolved in various concentrations of acetic acid (50%, 60%, 70%, and 80%) in an aqueous solution. In Figure 3, the electrospun fibers are shown. From the obtained micrographs, it can be observed that the different concentrations of acetic acid used in the solutions do not affect the quality of the fiber formation process. The electrospinning process was shown to be stable for all the solutions, with a uniform fiber formation, absent of any beads or particle formation.

As previously mentioned, varying the amount of acetic acid used to produce the fibers does not appear to affect the quality of the electrospun fibers. However, as shown in Table 3, the fiber diameter was influenced by the concentration of acetic acid used. From the measured diameters, it was noticed that the diameter of the fibers obtained increased as the concentration of acetic acid in the solution increased. Due to the variation in fiber diameter as a function of the acetic acid concentration, it is possible to electrospin fibers from the nano- to the microscale while maintaining a constant polymer content. The variation in the fibers’ diameter can be explained as follows. Chitosan is cationic polymer with amino groups attached to its backbone. In acidic solutions (with a pH of < 6), amino groups (—NH_2_) are protonated, forming —NH_3_ ions. Thus, they generate charge repulsions causing chitosan’s chain to expand. As the acetic acid content increases, the pH in the solution decreases, hence increasing the protonation of amino groups in chitosan. In addition, intermolecular hydrogen bonding allows the protonated amino groups to interact with the —OH groups of PVA [54,55]. The rheological properties of the solutions change as a result of the influence of acetic acid (a more detailed explanation is given in Section 3.1). The polymeric solution containing acetic acid at 50% had a lower viscosity and higher conductivity, producing fibers with a thinner diameter (~0.482 µm). On the other hand, fibers obtained from the solution containing acetic acid at 80% were thicker (~0.793 µm), as a result of its higher viscosity and lower conductivity. Similar results have been reported by Cheng T. et al. [28].

### 3.3. Thermogravimetric and Differential Scanning Calorimetry Analysis

As was previously mentioned, the aim of this paper is to examine how the electroactive properties of electrospun fibers can be affected by thermal treatments. Nonetheless, thermal treatments can also influence the thermal properties of the fibers. Chitosan/PVA fibers mats were thermally treated, by placing the samples in a drying oven for 24 h at t = 70 °C. Table 4 and Table 5 summarize the thermogravimetric analysis’s results. The thermogravimetric thermographs of non-thermally treated chitosan/PVA fibers exhibited a weight loss profile at three temperature stages, as shown in Figure 4a. The chitosan and PVA had weight losses at two stages. Regarding the PVA, the initial mass loss was in the temperature range of 51–134 °C, which can be attributed to the evaporation of the moisture content (~4%). Subsequently, a second weight reduction took place between 157 and 450 °C, indicating the thermal degradation of the PVA (~90.52%). In the case of the chitosan, the initial decrease in weight occurred within the temperature range of 35–118 °C, related to the evaporation of the moisture content (~5%). Subsequently, a second weight reduction is observed between 182 and 400 °C, indicating the thermal degradation and deacetylation of the chitosan (~49.68%).

Except for sample F-AA50, the mass loss of the non-thermally treated chitosan/PVA fibers happened in three stages [56]. The derivative thermogravimetric analysis curves showed that the first mass loss happens in a temperature range of 50–160 °C, associated with moisture and residual solvent evaporation. It is possible to observe from the DTG curves that, at the stage of the first mass loss, two peaks are formed. The presence of a second peak in this stage is due to acetic acid residues, which have a boiling point of 118 °C. The second mass loss was observed for F-AA50 in the range of 140–370 °C, and for samples F-AA60, F-AA70, and F-AA80 in the range of 180–370 °C. The second mass loss is related to the chitosan/PVA complex’s thermal destruction. A third mass loss was observable in samples F-AA60, F-AA70, and F-AA80 in the range of 375–500 °C, which could be related to the PVA byproducts and residuals of poly (vinyl acetate) degradation, which has a decomposition temperature of ~400 °C, present in PVA chains [57].

Regarding the samples that were thermally treated, significant changes were observed in their DTG curves, as shown in Figure 5b–e for the dried samples F-AA50, F-AA60, F-AA70, and F-AA80, respectively. Unlike the untreated samples, the thermally treated samples exhibited two mass loss stages. In the first mass loss stage, the thermally treated samples F-AA70 and F-AA80 showed the same two peaks (t = 60 °C and t = 125 °C) related to water and acetic acid residue evaporation. In contrast, the thermally treated sample F-AA60 did not display any peaks, and the thermally treated sample F-AA50 showed one peak related to water evaporation (t = 52 °C). Furthermore, for all the thermally treated samples, a peak shifting to a higher temperature in the second mass loss stage was observed. Sample F-AA50 exhibited the most noticeable shift, from 220 °C (for the non-thermally treated sample) to 231 °C (for the thermally treated sample). The obtained results are summarized in Table 5.

From the results obtained, it is noticeable that the thermal stability of the electrospun fibers decreased compared to the thermal stability of the pure chitosan and PVA (Table 4). Moreover, the thermal stability for samples F-AA60, F-AA70, and F-AA80 are relatively similar: 264, 264.8, and 266 °C, respectively. Sample F-AA50 shows a considerable decrease in its thermal stability to 220 °C in comparison to that of the other samples. Firstly, chitosan/PVA fibers have a lower thermal stability as a result of the polymer–polymer interaction. As a result, the structure of amorphous chitosan, which is dispersed along PVA chains, give rise to defects in the crystalline phase of PVA. This hinders the formation of crystalline regions. Therefore, the thermal energy required to break hydrogen bonds and melt free PVA chains is lower, thereby lowering the melting point of the chitosan/PVA.

Moreover, the TG results for the thermally treated electrospun fiber samples showed an increase in their thermal stability in comparison to those of the fibers samples that were not thermally treated. The increase in thermal stability can be attributed to the fact that the heat treatment facilitates the mobility of the polymer chains, allowing the amorphous regions of the chains to align and fold into crystalline regions [58,59]; hence, the thermal stability of the material increases as well. In addition, the increase in the thermal stability can be related to a physical crosslinking caused by the remotion of water molecules, allowing for higher intermolecular interactions.

The DSC thermographs for the chitosan and PVA are shown in Figure 6a. The chitosan’s thermograph showed an endothermic followed by an exothermic peak [60]. The broad endothermic peak found at 180 °C could be related to chitosan’s molecular chains’ arrangement. The exothermic peak found at 286 °C corresponded to chitosan’s thermal decomposition. The PVA DSC thermograph exhibited a shift in the baseline at 71 °C, which is consistent with the glass transition temperature (*tg*) for PVA [60]. Furthermore, an endothermic peak at 170 °C was observed, followed by a sharp endothermic peak at 220 °C, both of which are attributed to the melting point (*tm*) and PVA crystalline polymer fraction, respectively [61,62].

The DSC thermographs of the thermally and non-thermally treated chitosan/PVA fibers electrospun with different acetic acid concentrations are shown in Figure 6b–e. The DSC curves for all the samples (the thermally and non-thermally treated fibers) exhibited an endothermic peak followed by an exothermic peak. The endothermic peaks observed at 166, 166, 165, and 164 (F-AA50, F-AA60, F-AA70, and F-AA80, respectively) for the non-thermally treated fibers and 166, 164, 163.8, and 163 (F-AA50, F-AA60, F-AA70, and F-AA80, respectively) for the thermally treated fibers could be related to the melting point (*Tm*) of the chitosan–PVA blend. The exothermic peaks observed at 231, 232, 231, and 231 (F-AA50, F-AA60, F-AA70, and F-AA80, respectively) for the non-thermally treated fibers and 227, 231, 229, and 230 (F-AA50, F-AA60, F-AA70, and F-AA80, respectively) for thermally treated fibers could be associated with a crosslinking (complex formation between polymers) reaction on the chitosan molecules [63]. Table 6 summarizes the DSC values for the fiber samples.

The DSC curves of the non-thermally treated chitosan/PVA indicated that the baseline shift related to the PVA glass transition was no longer observed in any of the fiber samples. This is due to the shear stress caused by the electric field during electrospinning, which rearranges the polymer chains [64,65]. However, the DSC curves of the thermally treated chitosan/PVA fibers exhibited a change in the baseline at 37, 37, 37, and 39 °C (F-AA50, F-AA60, F-AA70, and F-AA80, respectively), as shown in Figure 6b–e. The decrease in *Tg* in the fiber composition could be related to the mutual interactions between the components and their partial compatibility [66,67]. However, there have been reports that electrospun fibers have higher crystallinity after the electrospinning process, as is the case with PVA fibers, as reported by Koosha et al. [63]. Numerous studies have reported that electrospun fibers experience a reduction in their crystalline structures as a result of the rapid solidification process of the stretched polymers. 

In summary, from the thermal analysis’s results, it is possible to notice that submitting the chitosan/PVA fibers to a thermal treatment at 70 °C for 24 h influenced the thermal properties of the material. The thermogravimetric curves corroborate that the thermal treatment promoted the elimination of water and solvent residues. The mass loss at the first stage, related to the solvent evaporation, decreased in all the thermally treated samples from 11.6 to 2.42%, 11.2 to 3.1%, 10.52 to 3.26%, and 9.58 to 6.47% for F-AA50, F-AA60, F-AA70, and F-AA80, respectively. Moreover, the DTG thermographs showed that the thermal stability of the electrospun fibers increased after been thermally treated. This can be explained as the result of a physical crosslinking effect. Another plausible explanation is that thermal treatments that use temperatures above the glass transition temperature of the polymers could provide enough energy to the polymer chains to move freely and rearrange, relaxing the stretched stress molecules after the electrospinning process, which is a prerequisite for sufficient crystallinity [58]. This was confirmed by the DSC analysis results, from which it was possible to calculate the degree of crystallinity (%) from the enthalpy of the melting point (∆H). As shown in Table 6, the degree of crystallinity of the electrospun fibers increased after being subjected to the thermal treatment, from 1.43 to 1.87%, 1.77 to 1.96%, 1.86 to 2.25%, and 1.30 to 1.90% for F-AA50, F-AA60, F-AA70, and F-AA80, respectively.

### 3.4. Tensile Properties of Electrospun Fibers

Table 7 displays the measured tensile strength, Young’s modulus, and elongation at break of the chitosan/PVA fiber mats, both before and after the thermal treatment. Based on the obtained results, a notable observation can be made regarding the significant alteration in the tensile properties of the electrospun polymeric fibers. These changes are evident not only in relation to the concentration of acetic acid but also as a direct outcome of the applied thermal treatment. Derived from the obtained results, it is noteworthy that fiber samples F-AA-50, F-AA-60, and F-AA-70 demonstrate distinctive curves that resemble those of hard/tough plastic polymers. These curves exhibit a distinct peak stress at the point of transition from elastic to plastic deformation, which is a characteristic feature of such materials. On the other hand, fiber sample F-AA80 shows the characteristic curve of a brittle polymer, which elastically deforms and fractures before deforming plastically, as shown in Figure 7. Other studies have reported the change in the mechanical properties of electrospun chitosan/PVA fibers in relation to the polymeric concentration. Zhou Y. et al. [67] evaluate the mechanical properties of electrospun chitosan/PVA fibers with different polymer concentrations. Their study showed that as the mass ratio of PVA/chitosan increased from 10/90 to 50/50, the tensile strength also increased from 2.78 to 5.55 MPa. Similar results were reported by Charernsriwilaiwat N. et al. [68], who investigated the influence of the polymer concentration on the mechanical properties of electrospun chitosan/PVA fibers. The tensile strength of chitosan/PVA fibers with mass ratios ranging from 10/90 to 50/50 decreased from 8.9 to 1.5 MPa, respectively. The tensile strength of pure PVA fibers was reported to be 12.8 MPa. Nonetheless, and to the best of the authors’ knowledge, there are not many works that report the influence of acetic acid concentration on the mechanical properties of electrospun fibers. Mohammad Z. et al. [69] reported the fabrication of chitosan fibers via extrusion and a coagulation bath. The fibers were obtained from solution containing 1.5 wt.% chitosan dissolved in different acetic acid concentrations (1, 2, 3, and 6 *v.%*). The mechanical test showed an improvement in the tensile strength in fibers from 0.7 to 1.2 MPa as the acetic acid increased from 1 to 2 *v.%*. Nevertheless, above 2 *v.%*, the tensile strength decreases to 0.3 MPa at 6 *v.%*. 

The change in the tensile properties of the electrospun chitosan/PVA fibers can be attributed to the plasticizing effect of the water and acetic acid molecules. The plasticizing effect of acetic acid has been reported by Carmiña et al. [70] and Zhang Y. et al. [71]. In their studies, they attributed the plasticizing effect to the large acetate ion which has a delocalized charge, capable of plasticizing the chitosan structure and facilitating a favorable long-range molecular arrangement, hence enhancing the mechanical properties. Similar results were reported by Vu T. et al. [72]. The authors investigated the mechanical properties of nanofibers prepared from an 8% PVA solution using electrospinning with varying concentrations of acetic acid (0%, 20%, 35%, and 50% *w/w*). The results indicated a significant rise in both the elongation at break and Young’s modulus with an increasing acetic acid concentration. Specifically, the elongation at break increased from 37% (acetic acid, 0% *w/w*) to 69.6% (acetic acid, 50% *w/w*), while the Young’s modulus exhibited an enhancement from 213 MPa (acetic acid, 0% *w/w*) to 449 MPa (acetic acid, 50% *w/w*).

Figure 7 shows the stress–strain diagrams of the thermally treated fiber mats. A decrease in both the tensile strength and elongation at break was observed for all the samples. The sample F-AA50 displayed a significant change in both the tensile strength and elongation at break, with values decreasing from 13.1 to 10.34 MPa and from 12.13 to 7.5%, respectively. In contrast, sample F-AA80 exhibited a lower change in both the tensile strength and elongation at break, with values decreasing from 11.43 to 9.56 MPa and from 5 to 4.3%, respectively. The change in the mechanical properties of the thermally treated fibers could be related to the evaporation of the acetic acid residues and water molecules. This could induce a non-covalent polymer–polymer interaction, primarily by the formation of hydrogen bonding [73], altering the tensile properties of the fiber mats. Furthermore, the thermal energy provided by the heat treatment can facilitate the mobility of polymer chains, allowing the amorphous regions to align and fold to form crystallites [53]. Many other works have reported the influence of thermal treatments on PVA fibers. Wong K. et al. [58] reported the change in mechanical properties measured for individual PVA fibers before and after 4 h of annealing at 135 °C. Their results shows that the mean elastic modulus of the individual fibers increased from 4.4 ± 1.4 GPa to 7.6 ± 2.3 GPa, exhibiting an increase of 80% over the fibers before the thermal treatment. Furthermore, Mahir Es-saheb et al. [74] assess the change in the mechanical properties of PVA nanofiber sheets after heating. In their work, the PVA nanofiber sheets were thermally treated at 85 and 140 °C. Their work shows that the yield stress changed from 6.981 MPa (for the non-heated fibers), 9.63 Mpa (the fibers heated at 85 °C), and 6.298 Mpa (the fibers heated at 140 °C). Meanwhile, the reported elongation at break changed from 59.81% (for the non-heated fibers), 31.26% (the fibers heated at 85 °C), and 29.71% (the fibers heated at 140 °C).

### 3.5. Electroactive Test Response

As can be seen from the results obtained in the previous sections of this manuscript, the physical properties of the electrospun fibers vary based on the parameters used before (the concentration of acetic acid for the polymer’s dissolution) and after (the thermal treatment) the fabrication of the fibers. Taking this into consideration, the aim of this section was to study the influence the thermal treatment and acetic acid concentration have on chitosan/PVA fibers’ electroactive response.

The electroactive response of the electrospun fibers was evaluated with an electrochemical cell via the measurement of the speed displacement of the fiber samples as a function of time under a cyclic potential between −10 and 10 V. The samples used in this experiment were obtained from the non-thermally and thermally treated electrospun fibers mats. The size of the samples used was 20 × 4 mm (length × width). The deformation response of the chitosan/PVA fiber sample under a cycling differential potential (−10 to 10 V) in an electrolytic solution (HCl with a pH of 3) is shown in Figure 8.

From the obtained results, it was observed that the different acetic acid concentrations used for the fabrication of the fibers had a strong influence on the electroactive response of the material, as shown in Figure 9a. Even though all the electrospun fibers had the same polymeric concentration, they should exhibit a similar deformation to an electric stimulus, as was observed in our previous work [39]. Nonetheless, it is possible to observe that as the concentration of acetic acid utilized for the production of the fibers rises, two interesting behaviors were exhibited by the fibers. Firstly, the samples that were electrospun from solutions with higher concentrations of acetic acid became less electrically responsive in acidic media (a pH of <7). Sample F-AA50 exhibited the largest and fastest electrical response of 0.77 mm s^−1^ at a pH of 3. Meanwhile, sample F-AA60 showed the fastest displacement of 0.546 mm s^−1^ at a pH of 4. On the other hand, F-AA80 was not suitable for use in acidic media below a pH of 5, due to the fact that all the samples dissolved after their immersion in the solutions. The second behavior worth mentioning is that all the samples exhibited a notable electrical sensitivity between a pH of 7 and 8, with a speed displacement peak in this region. In this case, the fibers that were electrospun from solutions with higher concentrations of acetic acid became more electrically responsive at a pH of 7–8. The highest speed displacement peak was found for samples F-AA60 and F-AA70, exhibiting a maximal speed displacement of 1.06 mm s^−1^ and 1.14 mm s^−1^ at a pH of 7, respectively. Meanwhile, samples F-AA50 and F-AA80 had the lowest speed displacement at a pH of 7: 0.75 mm s^−1^ and 0.51 mm s^−1^, respectively. From these results, it is possible to conclude that the electroactive response sensitivity of the fibers in acidic and basic media changes depending on the acetic acid concentration used. It was observed that the fibers that were electrospun from solutions with higher acid concentrations tend to have a lower electroactive response at a pH < 6, due to the high swelling and fast deterioration of the samples. These behaviors can be related to two factors. Firstly, the acetic acid remnants found in the fibers can promote the fast swelling and degradation of the samples fabricated with higher concentrations of acetic acid. Furthermore, as it was discussed in Section 3.2 (Study on the solutions’ rheological properties), the amount of —NH_2_ that can get ionized (NH^+^_3_) in the chitosan chain is directly proportional to the acid concentration used for its solubility. Therefore, the possible amount of free —NH_2_ groups and NH^+^_3_ cations could be higher in the fibers electrospun from highly concentrated acetic acid solutions. Thus, due to the increase in the amount of free —NH_2_ groups and cations, the chitosan/PVA fibers became more electroactive responsive at neutral and basic pHs. To the best of the authors’ knowledge, no similar results have been reported prior to the completion of this paper.

Figure 9b shows the speed displacement measured from the thermally treated samples. It is possible to notice that the thermal treatment has a strong influence on the fibers’ electroactive response. In comparison to the non-thermally treated samples, which had a fast deterioration in acidic media, the thermally treated samples showed a better stability and a higher deformation at an acidic pH (2–3). F-AA80 was the most remarkable case, as it went from not been appliable at a pH of 3 to exhibiting a speed displacement of 1.37 mm s^−1^ at a pH of 3. This improvement was also observable for all the samples, which increased their speed displacement from 0.77 mm s^−1^, 0.117 mm s^−1^, and 0.308 mm s^−1^ to 2.16 mm s^−1^, 1.56 mm s^−1^, and 1.38 mm s^−1^ at a pH of 3 for F-AA50, F-AA60, and F-AA70 (thermally treated), respectively. Furthermore, the fibers’ electroactive response that was exhibited in all the samples showed a similar trend. Two speed displacement peaks were exhibited, the first at a pH of 3 and the second at around a pH of 9–10. The fibers electrospun with higher concentrations of acetic acid showed a greater electrical response, having a maximal speed displacement of 1.56 mm s^−1^, 1.54 mm s^−1^, and 1.31 mm s^−1^ at a pH of 9 for F-AA60, F-AA70, and F-AA80 (thermally treated), respectively. The thermally treated fibers showed an improved electroactive response in comparison to the non-thermally treated fibers. The partial elimination of residual acetic acid and water, as well as the increase in fiber crystallinity, as observed in Section 3.4, may account for this improvement, such as their increased stability in acidic media. To the best of the authors’ knowledge, no similar results have been reported prior to the completion of this paper.

Briefly, we will explain the bending mechanism as a reaction to an electric stimulus exhibited by the fibers. The amino groups contained in chitosan are responsible for its electroactive properties. These —NH_2_ groups undergo protonation (NH^+^_3_) in an acidic environment with a pH below 7, thus causing the chitosan to function as a cationic polyelectrolyte [75]. As described in the literature, the bending deformation of the fibers under an electric field can be described using Flory’s theory of osmotic pressure [33,76]. By applying an electric field, free ions are attracted to their counter electrodes, creating a gradient concentration of mobile ions in the solution. Hence, this leads to a difference in osmotic pressure (∆π) between the anode side (π1) and the cathode side (π2), causing the fibers to bend [77,78].

It can be concluded that several factors significantly influence the electroactive response of materials based on the results obtained. Specifically, in the case of electrospun-fiber-based electroactive materials, certain parameters such as the solvent concentration used prior to electrospinning exhibit a substantial influence over the material’s electroactive properties. Additionally, physical treatments, particularly thermal treatments, have demonstrated a remarkable ability to profoundly alter the material’s electroactive response. Remarkably, it has also been observed that thermal treatments enhance the stability of the fibers in aqueous environments compared to non-thermally treated fibers. Considering the combined impact of these findings, it is evident that these parameters play a significant role in tailoring electroactive properties during the development of responsive materials.

### 3.6. Fourier-Transform Infrared Spectroscopy FTIR Analysis

The FTIR spectra of the chitosan, PVA powder, and chitosan/PVA nanofiber mats were analyzed to investigate the molecular interaction in the chitosan/PVA fibers. Figure 10 shows the FTIR spectra of the chitosan, PVA, and chitosan/PVA fibers (F-AA50, F-AA60, F-AA70, and F-AA80). Specific absorption peaks indicative of the molecular composition of chitosan were identified via a Fourier-transform infrared (FTIR) analysis. The peak observed at 3354 cm^−1^ could be related to the combined stretching vibrations of the O—H and N—H groups. The aliphatic C—H bonds that can be attributed to the peak at 2926 cm^−1^ indicate the stretching vibrations of aliphatic C—H bonds. Furthermore, the stretching vibration of the amino group is attributed to the peak at 1561 cm^−1^. Additionally, the saccharide structure of chitosan is associated with characteristic peaks at 892 and 1150 cm^−1^. These findings align with those of previous studies reported in [63,79].

The PVA’s FTIR spectrum shows distinctive absorption peaks that provide insights into its molecular characteristics. The hydroxyl group (—OH)’s stretching vibrations are associated with the absorption peak observed at approximately 3290 cm^−1^. The peak at 2937 cm^−1^ is attributed to antisymmetric stretching vibrations of the CH2 groups. The peaks observed at 1709 cm^−1^ indicate C=O bonds’ stretching vibrations present in the PVA acetate units. An absorption peak related to the vibration of the C—H bonds in the methyl group was observed at 1420 cm^−1^. The stretching of the C—O bonds associated with the crystalline portion of the polymer chain is observed at the absorption peak at 1141 cm^−1^. The spectrum of the C—O bond’s asymmetric stretching vibration in the acetate group was observed around 1087 cm^−1^. These findings are consistent with the existing literature in the field [80].

It is evident that the FTIR spectra are very similar to those of the PVA for all the chitosan/PVA fiber samples. A broad and intense band from 3000 to 3600 cm^−1^ was observed to be associated with O−H and N−H stretching vibrations. As a result of the dehydration process, this area is narrower and slightly sharper in thermally treated fiber mats [81]. The spectra reveal the formation of hydrogen bonds between PVA and chitosan, evident from the shift toward lower wavenumbers in the O−H and N−H stretching vibration peaks. Specifically, the peak at 3354 cm^−1^ for chitosan shifts to approximately 3300 cm^−1^ for all the chitosan/PVA fiber samples. The stretching vibrations of the C=O and C−O bonds of the acetate units in the PVA can be attributed to the peaks at 1712 cm^−1^ and 1640 cm^−1^. The thermally treated fibers showed a peak shifting from 1640 to 1652 cm^−1^ (associated with the C=O stretching in amide and amide I vibration), which could be due to the formation of an amide group from the reaction of carboxylic with amine groups, as a result of the heat treatment [81,82]. Additionally, a shift peak was observed from around 1590 cm^−1^ to 1562 cm^−1^, attributed to the hydrogen bonding between the PVA’s —OH groups and chitosan’s —NH group. The peaks at around 1410 (for the non-thermally treated fibers) and 1415 cm^−1^ (for the thermally treated fibers) are related to the methyl group (—CH_3_) C—H bond vibrations. The peak around 1075 cm^−1^ could be related to the C−O bond asymmetric stretching vibration of the acetate group. A peak associated with C—H bending vibrations in the molecule was found around 842 cm^−1^. These results are in good agreement with those of previous reports [63,83].

#### Spectra Deconvolution for the Determination of Intermolecular Hydrogen Bonding and Free Amine (—NH_2_) Variation Post-Thermal Treatment

It was found that the mechanical properties, the thermal properties, and the electroactive properties of the electrospun chitosan/PVA fibers were altered after being subjected to the thermal treatment. The change in the thermal and tensile properties, as was established previously, could be related to two possible effects: the increase in the crystallinity and/or the physical crosslinking between polymers chains. Nonetheless, the FT-IR spectra (Figure 10) showed that the characteristic peak associated with the crystalline part of the PVA (1141 cm^−1^) observed in the PVA spectrum overlapped the peak at ~1075 cm^−1^ for all the fiber mat samples. Moreover, after being thermally treated, the fiber samples did not exhibit any noticeable change at this range. Thus, the change in the mechanical and thermal properties could be strongly related to a physical crosslinking effect, as a result of the removal of acetic acid and water residues from the microstructure of the electrospun fibers. The remotion of the acetic acid and water residues from the microstructure could lead to an increase in the intermolecular hydrogen bonding between the —NH_2_ and —OH groups of the chitosan with the —OH groups of the PVA. On the other hand, from the electroactive response test, it was reported that the fibers electrospun from the solutions with high acetic acid concentrations at a pH below 5 exhibited a low to non-tip displacement. Moreover, these samples were dissolved after being immersed in the acidic medium. However, after being thermally treated, not only were their electroactive properties drastically enhanced, as shown in Figure 9, but the solubility of fiber mats was also reduced, up to the point that all the samples showed a high stability in acidic media. The higher stability of the fibers in the aqueous media could also due to the physical crosslinking effect caused by the thermal treatment. Meanwhile, the improvement in their electroactive response can be associated with an increase in the proportion of the free amino groups anchored to the molecular structure of the fibers. 

As a consequence of the thermal treatment, the fibers present a change in their physical properties, which could be due to the variation in the intermolecular hydrogen bonding between the polymers and the variation in the free amino groups. In order to elucidate the variation in the proportion of hydrogen bonding interactions and free amine, after the thermal treatment, a deconvolution analysis employing Gaussian line shapes was employed on the peak of the Fourier-transform infrared (FTIR) spectrum in the range of 3000–3700 cm^−1^. The —OH and the —NH region were studied in order to analyze the types of hydrogen bonds. In the —NH region, the free amine absorption peak is 3408 cm^−1^; the intermolecular association (N_2—_H_1_…O_5_/N_2_—H_2_…O_1_) peak is around 3335 cm^−1^; the intramolecular association (O_3_H…O_5_/O_3_H…O_6_) absorption peak is around 3366 cm^−1^; the amide group (—CONH-) absorption peak is around 3240 cm^−1^; and the primary ammonium (—NH^+^_3_) absorption peak is around 3100 cm^−1^. In the −OH region, the free hydroxyl (—OH) absorption peak is around 3580 cm^−1^, and the multimer intermolecular association (O_6_H…N_2_) peak is around 3462 cm^−1^ [84,85,86,87,88]. Table 8 shows the results obtained from the spectra deconvolution.

From Table 8, it is noticeable that the fibers that were thermally treated showed a variation in relative strength (%) at the peak around 3408 cm^−1^, associated with the free amine. The proportion of —NH_2_ for all the samples increased after the thermal treatment, and sample F-AA50 had the highest increment from 6.39% to 6.9%. This variation can explain the change in the electroactive properties of the fibers. Furthermore, the peaks associated with the intermolecular hydrogen bonding exhibited an increase in all the samples that were thermally treated. The peak at 3335 cm^−1^ associated with intermolecular association (N_2—_H_1_…O_5_/N_2_—H_2_…O_1_) increased from 27.5%, 27.46%, 27.7%, and 27.6% to 28%, 29.15%, 27.9%, and 28.9% for F-AA50, F-AA60, F-AA70, and F-AA80, respectively. The peak at 3462 cm^−1^ associated with multimer-intermolecular association (O_6_H…N_2_) increased from 22.5%, 20%, 23.3%, and 21.77% to 22.5%, 21.2%, 23.35%, and 21.77%, for F-AA50, F-AA60, F-AA70, and F-AA80, respectively. Additionally, it was observed that the proportion in the peak related to primary ammonium (—NH^+^_3_) at 3100 cm^−1^ decreased in the samples that were thermally treated. From these results, it can be deduced that the thermal treatment induces a physical crosslinking effect on the chitosan/PVA fibers.

## 4. Conclusions

In conclusion, this study successfully electrospun micro- and nano-scale fibers from chitosan/PVA solutions with varying concentrations of acetic acid (50, 60, 70, and 80 v/v%). The rheological properties of the polymeric solutions were notably affected by the acetic acid concentration. An increased acetic acid concentration led to a higher viscosity (from 6157 mPa·s for AA50 to 8747.3 mPa·s for AA80) and reduced conductivity (from 2852.6 µS cm^−1^ to 999.6 cm^−1^ for AA50 and AA80, respectively). The fiber morphology correlated with the rheological properties, with the fiber diameter increasing as the acetic acid concentration rose, ranging from 0.482 to 0.793 µm. The thermal treatment at 70 °C enhanced the fiber thermal stability, as shown by the thermogravimetric analysis (TG). The mechanical properties were influenced by the acetic acid concentration. The fiber samples with 50%, 60%, and 70% acetic acid concentrations exhibited curves resembling hard/tough plastic polymers, while the sample with 80% acetic acid demonstrated the characteristics of a brittle polymer. The observed changes in mechanical properties were attributed to the plasticizing effect of the large acetate ion. Electroactive properties were also assessed. The untreated fibers showed a decreased electroactive response in more acidic environments as the acetic acid concentration increased. The fibers electrospun from solutions with 50% acetic acid reached a maximum displacement of 0.7728 mm s^−1^ at a pH of 3, while those electrospun from solutions with 80% acetic acid dissolved at the same pH. The thermally treated fibers exhibited an improved electroactive response, with a heightened sensitivity to basic mediums (pH > 7). All the samples exhibited two displacement peaks, one at a pH of 3 and another around a pH of 9–10. The FT-IR spectra revealed increased intermolecular hydrogen bonding due to the thermal treatment. This could be related to the higher intermolecular polymer chain interaction as a result of a physical crosslinking effect induced by the thermal treatment. Overall, this study provides insights into how acetic acid concentration and thermal treatment impact the mechanical, thermal, and electroactive properties of chitosan/PVA fibers. These findings advance our understanding of the development of electro-responsive soft actuators. Additionally, comprehending how pre- and post-fabrication parameters impact the electroactive properties of a material allows for the development of soft actuators with tunable electroactive responses without altering the material’s composition. This approach has promising implications, particularly in the field of medical applications, in which the development of adaptive soft grippers could revolutionize minimally invasive procedures.

## Figures and Tables

**Figure 1 polymers-15-03719-f001:**
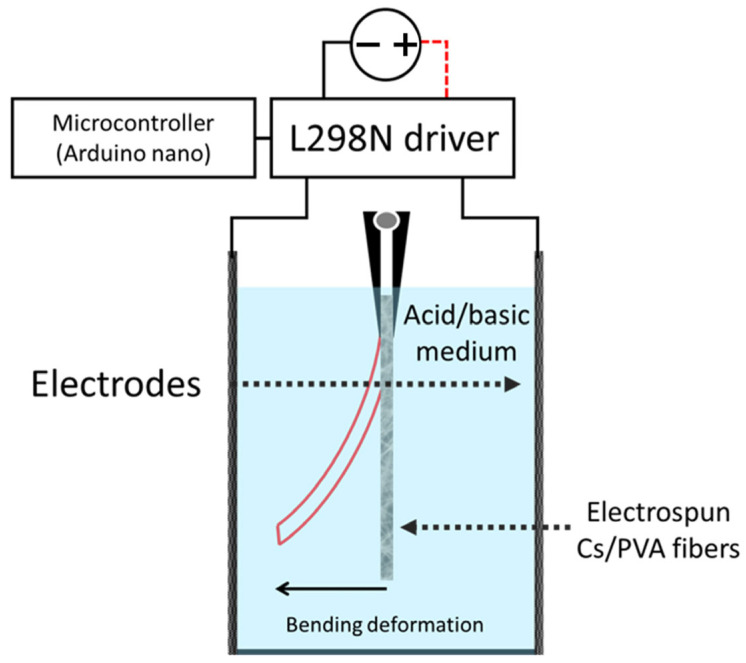
Schematic showing how the electroactive response of fiber samples was evaluated.

**Figure 2 polymers-15-03719-f002:**
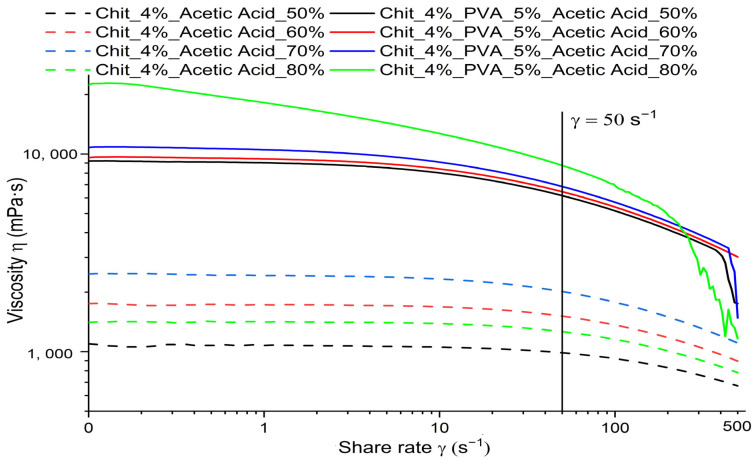
Scheme of dependency of viscosity with respect to the concentration of acetic acid in polymer solutions.

**Figure 3 polymers-15-03719-f003:**
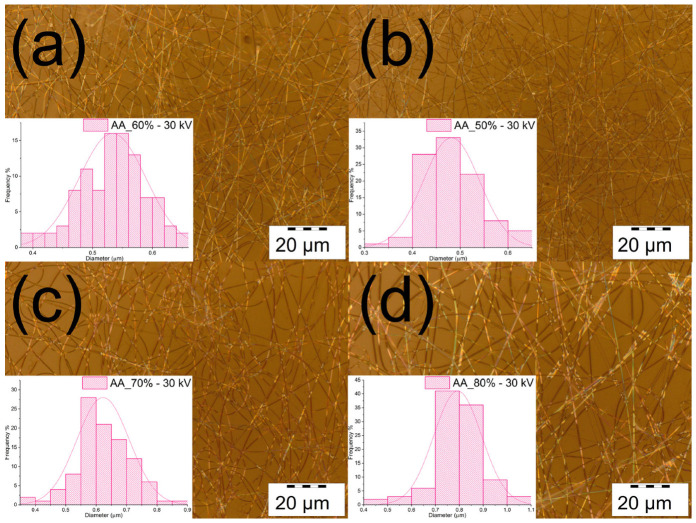
Microscope images of the chitosan/PVA electrospun fibers. (**a**) Chitosan 4 wt.%/PVA 5 wt.% dissolved in 50% acetic acid; (**b**) chitosan 4 wt.%/PVA 5 wt.% dissolved in 60% acetic acid; (**c**) chitosan 4 wt.%/PVA 5 wt.% dissolved in 70% acetic acid; and (**d**) chitosan 4 wt.%/PVA 5 wt.% dissolved in 80% acetic acid.

**Figure 4 polymers-15-03719-f004:**
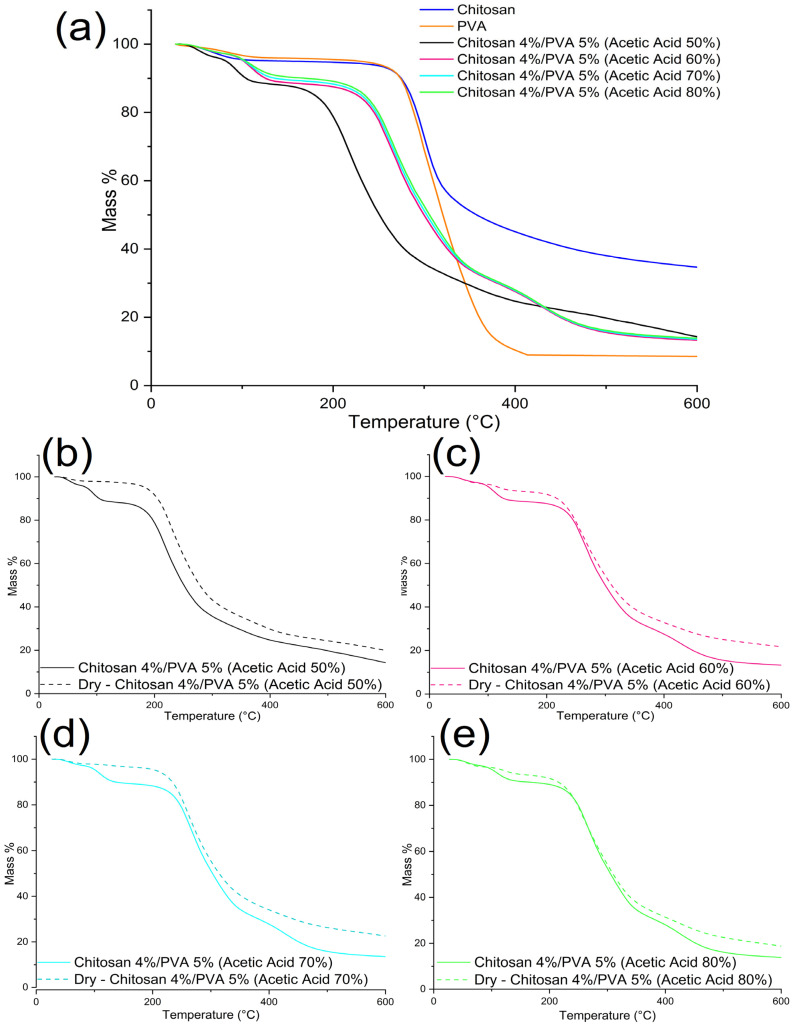
Thermogravimetric thermographs of (**a**) chitosan, PVA, and chitosan/PVA nanofibers, electrospun from polymeric solutions with different acetic acid contents. (**b**) Chitosan 4%/PVA 5% dissolved in 50% acetic acid; (**c**) chitosan 4%/PVA 5% dissolved in 60% acetic acid; (**d**) chitosan 4%/PVA 5% dissolved in 70% acetic acid; and (**e**) chitosan 4%/PVA 5% dissolved in 80% acetic acid.

**Figure 5 polymers-15-03719-f005:**
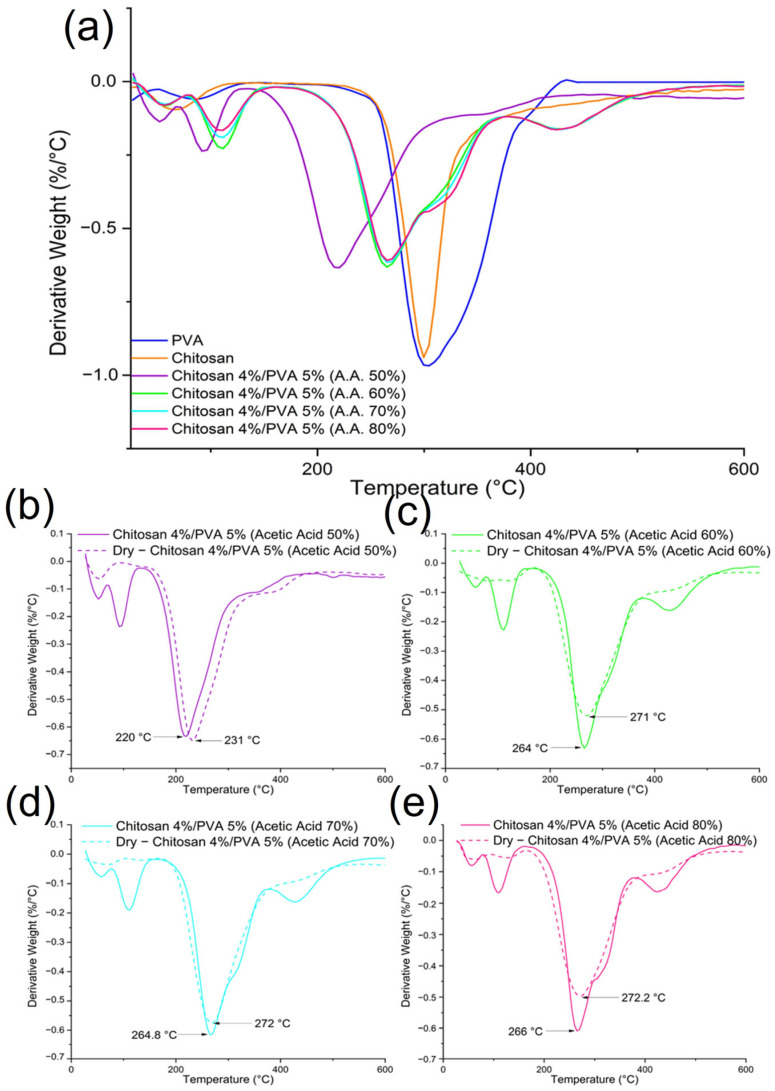
DTG thermographs of (**a**) chitosan, PVA, and chitosan/PVA fibers, electrospun from polymeric solutions with different acetic acid contents. (**b**) Chitosan 4%/PVA 5% dissolved in 50% acetic acid; (**c**) chitosan 4%/PVA 5% dissolved in 60% acetic acid; (**d**) chitosan 4%/PVA 5% dissolved in 70% acetic acid; and (**e**) chitosan 4%/PVA 5% dissolved in 80% acetic acid.

**Figure 6 polymers-15-03719-f006:**
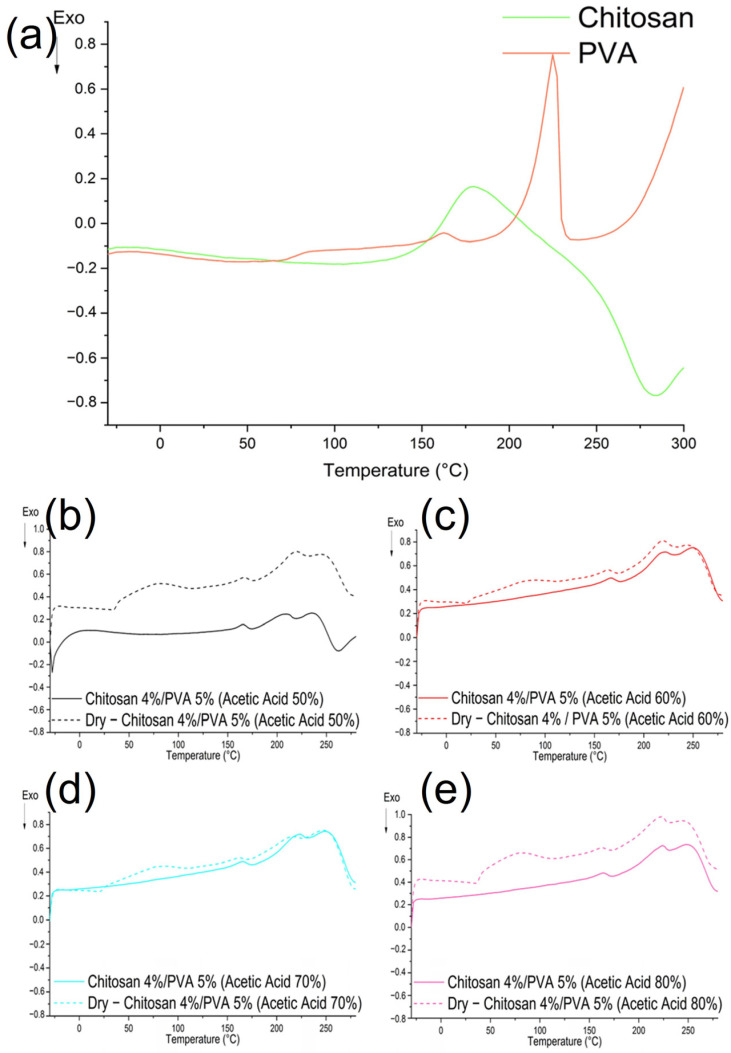
DSC curves of (**a**) chitosan, PVA, and chitosan/PVA fibers, electrospun from polymeric solutions with different acetic acid content. (**b**) Chitosan 4%/PVA 5% dissolved in 50% acetic acid; (**c**) chitosan 4%/PVA 5% dissolved in 60% acetic acid; (**d**) chitosan 4%/PVA 5% dissolved in 70% acetic acid; and (**e**) chitosan 4%/PVA 5% dissolved in 80% acetic acid.

**Figure 7 polymers-15-03719-f007:**
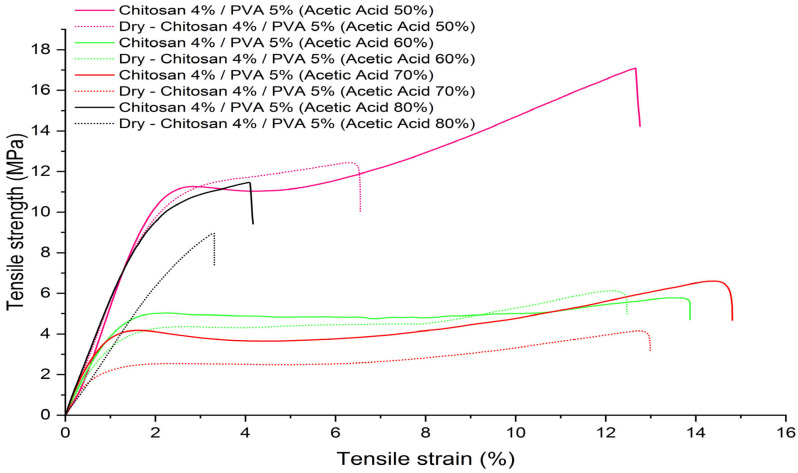
Stress–strain diagrams of chitosan/PVA fibers samples before and after heat treatment.

**Figure 8 polymers-15-03719-f008:**
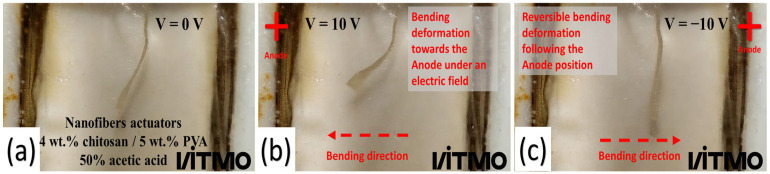
Bending deformation of chitosan/PVA nanofiber, eletrospun from the polymeric solution containing acetic acid at 50% under an applied DC voltage. (**a**) The applied electrical potential was equal to 0 V; (**b**) the applied electrical potential was equal to 10 V; and (**c**) the applied electrical potential was equal to −10 V (the electrode’s polarities were inversed).

**Figure 9 polymers-15-03719-f009:**
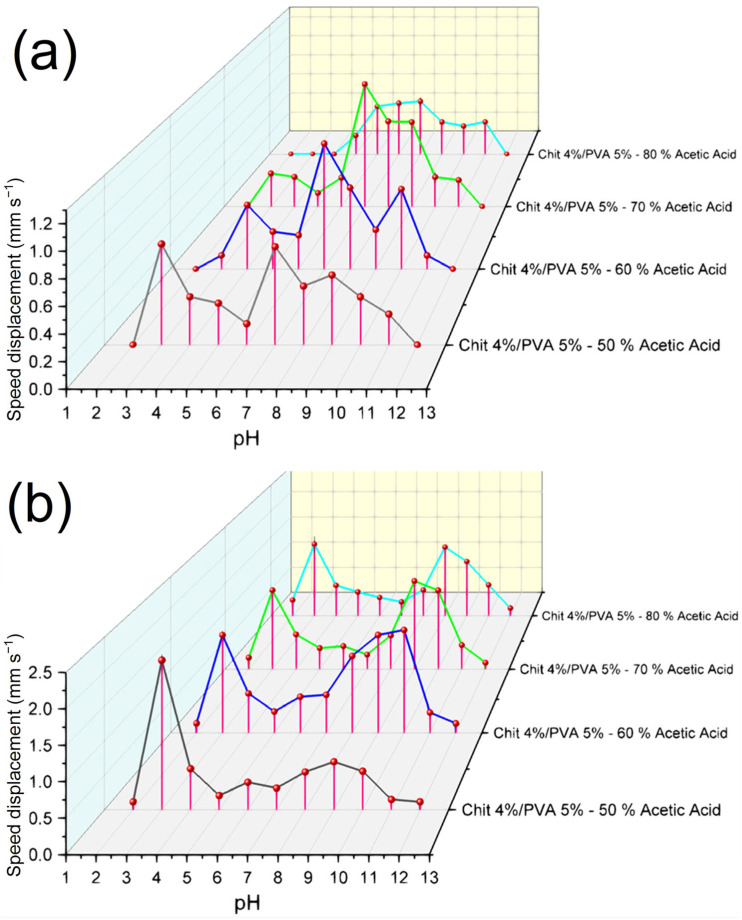
Speed displacement of (**a**) non-thermally treated chitosan/PVA fibers; and (**b**) thermally treated chitosan/PVA fibers.

**Figure 10 polymers-15-03719-f010:**
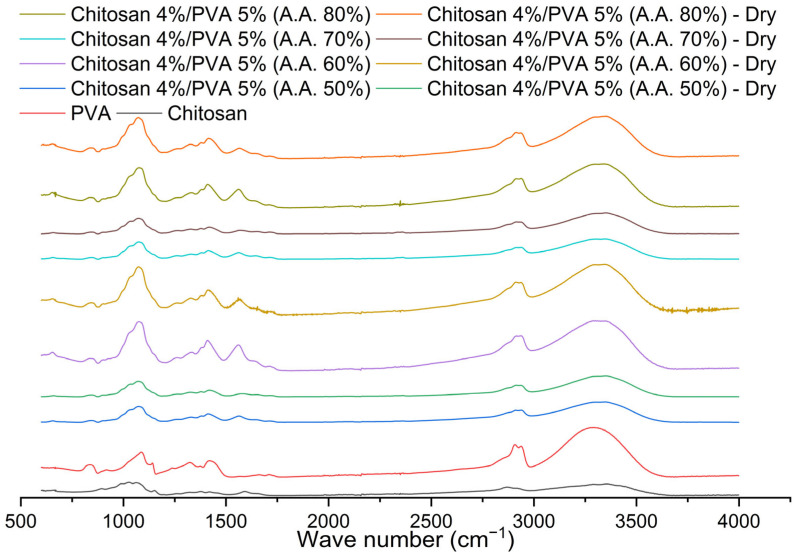
Spectra observed for chitosan, PVA, and chitosan/PVA fibers samples.

**Table 1 polymers-15-03719-t001:** Apparent viscosity measured at $\dot{\gamma }$ = 50 s^−1^ of chitosan and chitosan/PVA solutions dissolved in different acetic acid concentrations.

Acetic Acid Concentration	50%		60%		70%		80%	
	Cs.	Cs/PVA	Cs.	Cs/PVA	Cs.	Cs/PVA	Cs.	Cs/PVA
Viscosity (η) [mPa·s]	986.93	6157	1512.5	6445	2014.2	6847.7	1262.8	8747.3

**Table 2 polymers-15-03719-t002:** Conductivity of chitosan and chitosan/PVA solutions dissolved in different concentrations of acetic acid.

Acetic Acid Concentration	50%		60%		70%		80%	
	Cs.	Cs/PVA	Cs.	Cs/PVA	Cs.	Cs/PVA	Cs.	Cs/PVA
Conductivity (µS cm^−1^)	3558.4	2852.6	2695.2	2126.2	2395.4	1682.6	1245.6	999.6

**Table 3 polymers-15-03719-t003:** Mean diameter of chitosan/PVA fibers electrospun from solutions with different acetic acid contents.

Sample	Chitosan (wt.%)	PVA (wt.%)	Acetic Acid (%)	Mean (µm)	Morphology
AA50	4	5	50	0.482 ± 0.006	Uniform fiber formation
AA60	4	5	60	0.533 ± 0.01	Uniform fiber formation
AA70	4	5	70	0.622 ± 0.0059	Uniform fiber formation
AA80	4	5	80	0.793 ± 0.011	Uniform fiber formation

**Table 4 polymers-15-03719-t004:** Thermogravimetric analysis of chitosan powder, PVA powder, and chitosan/PVA fibers (not thermally treated).

Sample	Chitosan (wt.%)	PVA (wt.%)	First Mass Loss (%) 50–160 °C	Second Mass Loss (%) 175–370 °C	Third Mass Loss (%) 375–500 °C	First Stage 1st Peak (°C)	First Stage 2nd Peak (°C)	Second Stage (°C)	Third Stage (°C)
PVA			4	90.53		81		304	
Chit			5	49.68		67		299	
F-AA50	4	5	11.6	65.98		51	93	220	
F-AA60	4	5	11.2	58.18	15.86	56	100	264	428
F-AA70	4	5	10.52	57.98	16.44	56	109	264.8	428
F-AA80	4	5	9.58	59.27	15.81	56	108	266	427

**Table 5 polymers-15-03719-t005:** Thermogravimetric analysis of chitosan/PVA fibers (thermally treated). Samples were dried at 70 °C for 24 h.

Sample	Chitosan (wt.%)	PVA (wt.%)	First Mass Loss (%) 50–160 °C	Second Mass Loss (%) 140–370 °C	Third Mass Loss (%) 375–500 °C	First Stage 1st Peak (°C)	First Stage 2nd Peak (°C)	Second Stage (°C)	Third Stage (°C)
F-AA50	4	5	2.42	71.07		52		231	
F-AA60	4	5	3.1	71.62		60		271	
F-AA70	4	5	3.26	71.35		60	125	272	
F-AA80	4	5	6.47	71.79		63	125	275	

**Table 6 polymers-15-03719-t006:** Glass transition temperature (*Tg*), melting temperature (*Tm*), and degree of crystallinity (%) of chitosan/PVA nanofibers samples before and after heat treatment.

Sample	Glass Transition *t(g)*		Melting Point *t(m)*		Degree of Crystallinity (%)	
	Not Dry	Dry	Not Dry	Dry	Not Dry	Dry
F-AA50		37	164.2	165.15	1.43	1.87
F-AA60		37	166.0	166.4	1.77	1.96
F-AA70		37	165.5	166.8	1.86	2.25
F-AA80		39	164.45	164.97	1.30	1.90

**Table 7 polymers-15-03719-t007:** Young’s modulus, tensile strength, and elongation at break of chitosan/PVA nanofibers samples before and after heat treatment.

Sample	Young’s Modulus (MPa)		Tensile Strength (MPa)		Elongation at Break (%)	
	Not Dry	Dry	Not Dry	Dry	Not Dry	Dry
F-AA50	649.19 ± 52.3	592.3 ± 50.4	13.1 ± 1.22	10.34 ± 1.5	12.13 ± 1.2	7.5 ± 1.21
F-AA60	474.8 ± 23.8	462.6 ± 28.25	5.8 ± 1.1	4.11 ± 1.01	13.68 ± 2.9	11.6 ± 3.19
F-AA70	594.1 ± 53.6	472.1 ± 23.7	5.26 ± 0.69	4.6 ± 0.8	15.06 ± 0.36	14.3 ± 1.5
F-AA80	545.92 ± 52.9	375.415 ± 80.4	11.43 ± 1	9.56 ± 2.6	5 ± 0.86	4.3 ± 0.74

**Table 8 polymers-15-03719-t008:** Relative strength (%) of the deconvoluted bands for thermally and non-thermally treated chitosan/PVA fiber samples.

Sample	Types of Hydrogen Bond		Abbreviation	Wave Number (cm^−1^)	Relative Strength (%)	Relative Strength/% (Dried Samples)
Acetic acid 50%	Primary ammonium	I	—NH^+^_3_	~3100 cm^−1^	3.93	3.2
	Intermolecular hydrogen bond	II	OH…ether O	~3200 cm^−1^	8.7	5.2
	Amide	III	—CONH—	~3240 cm^−1^	30.1	33
	Intermolecular association	IV	N_2_—H_1_…O_5_/N_2_—H_2_…O_1_	~3335 cm^−1^	27.5	28
	Intramolecular association	V	O_3_H…O_5_/O_3_H…O_6_	~3366 cm^−1^	0.8	0.9
	Free amine	VI	—NH_2_	~3408 cm^−1^	6.39	6.9
	Multimer (Intermolecular association)	VII	O_6_H…N_2_	~3462 cm^−1^	22.13	22.6
	Free hydroxyl	VIII	—OH	~3580 cm^−1^	0.36	0.43
Acetic acid 60%	Primary ammonium	I	—NH^+^_3_	~3100 cm^−1^	4.4	4.2
	Intermolecular hydrogen bond	II	OH…ether O	~3200 cm^−1^	10.27	8.2
	Amide	III	—CONH—	~3240 cm^−1^	029.3	30.2
	Intermolecular association	IV	N_2_—H_1_…O_5_/N_2_—H_2_…O_1_	~3335 cm^−1^	27.46	29.15
	Intramolecular association	V	O_3_H…O_5_/O_3_H…O_6_	~3366 cm^−1^	0.90	0.92
	Free amine	VI	—NH_2_	~3408 cm^−1^	6.14	6.45
	Multimer (Intermolecular association)	VII	O_6_H…N_2_	~3462 cm^−1^	20	21.1
	Free hydroxyl	VIII	—OH	~3580 cm^−1^	0.09	0.21
Acetic acid pse-70%	Primary ammonium	I	—NH^+^_3_	~3100 cm^−1^	4.21	3.66
	Intermolecular hydrogen bond	II	OH…ether O	~3200 cm^−1^	9.6	6.69
	Amide	III	—CONH—	~3240 cm^−1^	29.7	32.1
	Intermolecular association	IV	N_2_—H_1_…O_5_/N_2_—H_2_…O_1_	~3335 cm^−1^	26.6	27.9
	Intramolecular association	V	O_3_H…O_5_/O_3_H…O_6_	~3366 cm^−1^	0.88	0.91
	Free amine	VI	—NH_2_	~3408 cm^−1^	5.89	5.96
	Multimer (Intermolecular association)	VII	O_6_H…N_2_	~3462 cm^−1^	21.32	23.35
	Free hydroxyl	VIII	—OH	~3580 cm^−1^	0.4	0.64
Acetic acid 80%	Primary ammonium	I	—NH^+^_3_	~3100 cm^−1^	4.5	4.0
	Intermolecular hydrogen bond	II	OH…ether O	~3200 cm^−1^	10.53	6.23
	Amide	III	—CONH—	~3240 cm^−1^	27	32.44
	Intermolecular association	IV	N_2_—H_1_…O_5_/N_2_—H_2_…O_1_	~3335 cm^−1^	27.6	28.9
	Intramolecular association	V	O_3_H…O_5_/O_3_H…O_6_	~3366 cm^−1^	0.71	0.79
	Free amine	VI	—NH_2_	~3408 cm^−1^	6.56	6.85
	Multimer (Intermolecular association)	VII	O_6_H…N_2_	~3462 cm^−1^	21.08	21.77
	Free hydroxyl	VIII	—OH	~3580 cm^−1^	0.1	0.27

## Data Availability

The data presented in this study are available on request from the corresponding author.

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
