# Peer review of "Influence of Thermal Treatment and Acetic Acid Concentration on the Electroactive Properties of Chitosan/PVA-Based Micro- and Nanofibers"

_polymers, 2023, doi:10.3390/polym15183719_

Round 1
Reviewer 1 Report
This manuscript needs refinement in several places. Detailed comments below.
Line 123 - 1132: This section is unnecessary. A detailed description of the research can be found in the methodology. This is also not the place for major observations or research conclusions.
Line 145: Samples mixed by hand or mechanical/electromagnetic stirrer used?
Line 149: What heater was used in the research? Was it possible to keep the temperature exactly 90°C? Of course, this also applies to other temperatures.
Line 151: On what basis did you choose such proportions of concentrations and the temperature of sample preparation?
Line 166: Add a more detailed description of the meter parameters. In addition (device model: manufacturer, city, country). Review the methodology and complete the missing descriptions.
Line 216: Enter micrometer accuracy.
Line 227: Why samples were dried at 70 °C. Did this temperature affect the quality of the final samples?
Line 487: Are your results close or far from what you expected. Compare the obtained results to similar materials described in the literature? This generally applies to all results obtained.
Figure 10: These photos are virtually the same. You have to indicate the differences visible in the photo, e.g. with arrows?
Line 694: In the description of some studies there is a poor discussion of research results. (eg mechanical properties) Too few references to scientific literature. This should be supplemented.
Line 730: Add a forward-looking conclusion, a forward-looking conclusion.
Reviewer 2 Report
In this manuscript, the author investigated the influence of thermal treatment and acetic acid concentration on the electroactive properties of chitosan/PVA-based micro and nano fibers. Some issues need addressed list as follow:
(1) What is electroactive properties? Which should introduced in the section of Introduction.
(2) Does the voltage has a influence on the electroactive properties?
(3) The thermal treatment procedure should introduced in the section of section 2. Materials and Methods.
(4) The format of tables and captions should revised according to the template.
(5) Why does the viscosity of Cs. Decreased when the acetic concentration reached to 80%?
(6) As the data in Figure 3 and table 2 are the same, one of which should deleted.
(7) The quality of Figure 6 should improved.
(8) What is the meaning for the TG investigation of chitosan/PVA and dry-chitoan/PVA (Figure 6 b-e)? the difference of TG investigation of chitosan/PVA and dry-chitoan/PVA should discussed in the manuscript. As well as figure 7 and 8.
Moderate editing of English language required
Round 2
Reviewer 2 Report
All of the issues mentioned were resolved in detail
Minor editing of English language required